# Feasibility and Tolerability of a Culture-Based Virtual Reality (VR) Training Program in Patients with Mild Cognitive Impairment: A Randomized Controlled Pilot Study

**DOI:** 10.3390/ijerph17093030

**Published:** 2020-04-27

**Authors:** Jong-Hwan Park, Yung Liao, Du-Ri Kim, Seunghwan Song, Jun Ho Lim, Hyuntae Park, Yeanhwa Lee, Kyung Won Park

**Affiliations:** 1Health Convergence Medicine Laboratory, Biomedical Research Institute, Pusan National University Hospital, Busan 49241, Korea; parkj@pusan.ac.kr (J.-H.P.); drkim4100@gmail.com (D.-R.K.); 2Department of Health Promotion and Health Education, National Taiwan Normal University, Taipei 10610, Taiwan; liaoyung@ntnu.edu.tw; 3Department of Thoracic and Cardiovascular Surgery, Pusan National University School of Medicine, Biomedical Research Institute, Pusan National University Hospital, Busan 49241, Korea; song77.sh@gmail.com; 4Department of Neurology, College of Medicine, Dong-A University, Busan 49201, Korea; tortoises84@naver.com; 5Department of Health Care and Science, Dong-A University, Busan 49315, Korea; htpark@dau.ac.kr; 6SY Inotech, Busan 48520, Korea; robocia@naver.com; 7Busan Metropolitan Dementia Center, Busan 49201, Korea

**Keywords:** older adults, cognitive function, intervention, dementia

## Abstract

The present study examined whether a culture-based virtual reality (VR) training program is feasible and tolerable for patients with amnestic mild cognitive impairment (aMCI), and whether it could improve cognitive function in these patients. Twenty-one outpatients with aMCI were randomized to either the VR-based training group or the control group in a 1:1 ratio. The VR-based training group participated in training for 30 min/day, two days/week, for three months (24 times). The VR-based program was designed based on Korean traditional culture and used attention, processing speed, executive function and memory conditions to stimulate cognitive function. The adherence to the culture-based VR training program was 91.55% ± 6.41% in the VR group. The only adverse events observed in the VR group were dizziness (4.2%) and fatigue (8.3%). Analysis revealed that the VR-based training group exhibited no significant differences following the three-month VR program in Korean Mini-Mental State Examination (K-MMSE) scores, working memory functions such as performance on the digit span test, or in Stroop test performance and word fluency. We conclude that although the 12-week culture-based VR training program did not improve cognitive function, our findings revealed that the culture-based VR training program was feasible and tolerable for participants with aMCI.

## 1. Introduction

Amnestic mild cognitive impairment (aMCI) is a condition of declined cognitive function that might indicate the early phases of Alzheimer’s disease (AD) or other dementias [1]. Patients with aMCI are considered to be in the stage between normal aging and dementia, and still have the ability to perform daily errands [1,2]. Therefore, early intervention at the aMCI stage can allow patients to retain and even improve their cognitive function [3].

Innovative technologies have recently played an important role in cognitive assessment, cognitive modeling, and changes in the field of neuropsychology [4,5]. In particular, virtual reality (VR) is a tool that allows subjects to be completely involved in an imitation of an actual situation, which is rich in temporal naturalistic and spatial contexts. VR helps to address a number of limitations in the research of cognitive function due to its distinct characteristics [6]. It is noted that VR shares the same basic mechanism with the human brain—embodied simulation—and thus with this embodied technology it is possible to facilitate cognitive modeling and change by designing virtual environments to simulate both the external and internal world [7]. Therefore, VR has emerged as a rising tool of cognitive behavioral therapy and cognitive rehabilitation for aMCI and dementia [8]. Previous studies reported that performance in VR applications has been shown to correlate with subjective memory complaints [9], and VR applications have exhibited promising results during the assessment of cognitive functions [9,10]. A feasibility study targeting healthy older adults showed that participants were motivated by VR-based tasks and environment, and that these tasks and environments could be helpful to improve the participants’ memory in a laboratory setting [11].

However, previous studies are limited in several aspects. Firstly, a majority of previous VR studies has been conducted in Western countries, and limited studies have been conducted in the context of culture-based VR. In particular, culture-based VR training programs for subjects of different cultural or environmental backgrounds are still needed. For example, a program can be designed based on a Korean traditional game, which may spur older Korean adults’ memory and increase their motivation to attend a VR training program. Secondly, previous studies were also limited by only focusing on memory. It remains unclear what the effects of VR-based training programs are on different domains of cognitive function, such as attention, executive function and processing speed. Finally, most previous studies were conducted in community-based settings [11], which may be limited by a lack of professionals to perform neuropsychological assessments. To fill these research gaps, the present study used a clinical setting and aimed to examine: (1) whether a culture-based VR training program is feasible and tolerable for patients with aMCI; and (2) the effect of a 12-week culture-based VR training program on different domains of cognitive function among patients with aMCI.

## 2. Materials and Methods

### 2.1. Participants

Patients aged between 50 and 80 years with a diagnosis of amnestic mild cognitive impairment (aMCI) were eligible in the present study. Based on the operationalized Petersen aMCI criteria [1], the inclusion criteria were as follows:Normal mental status: A MMSE score of more than 1.5 Standard Deviations (SD) below age- and education-adjusted normative means [12,13].Memory complaint: A subjective memory complaint that was confirmed by an informant.Impaired memory: A delayed recall score on the Seoul Verbal Learning test < 1 SD below age- and education-adjusted normative means [14,15].Not demented: A Clinical Dementia Rating (CDR) scale of <0.5 [16].Normal daily functioning: the Seoul Instrumental Activities of Daily Living (IADL) ≤ 7 [17].

All subjects were not diagnosed with dementia, had a Hachinski Ischemic Score of ≤4 [18], and a brain magnetic resonance imaging (MRI) or computed tomography (CT) scan showing no other condition that could cause cognitive impairment.

The exclusion criteria included the following: (1) The subject had a history of stroke or was being treated for a stroke or epilepsy; (2) the subject had other suspected degenerative diseases or mental illnesses; and (3), subjects who were depressed or abusing substances, had head injuries, thyroid malfunction or other medical abnormalities that could impair cognitive function. We conducted this study according to the International Harmonization Conference guidelines on Good Clinical Practice. The procedure of this study was reviewed by the Institutional Review Board of Dong-A University Hospital. All subjects or their legally authorized representatives provided written informed consent for the participation in the study. Finally, subjects were 21 out-patients with aMCI, who were randomized to either the VR-based training group or the control group. This study was registered in the University Hospital Medical Information Network (UMIN) Clinical Trials Registry (no. UMIN000030187).

### 2.2. Study Design

The patients who had been diagnosed with aMCI by a dementia clinician from a memory and dementia clinic at a university hospital were recruited. We used Statistical Analysis System (SAS) programming to randomly assign the participants with a 1:1 ratio to a VR group (*n* = 10) and a control group (*n* = 11) by the block randomization method. The VR-based training program was designed as a multicomponent restorative cognitive function training. Participants in the control group were asked to maintain their normal daily activities during the study, and were informed that they would be able to participate in the virtual reality (VR)-based training program after the end of the study. Assessments were performed before the start of the VR-based training program (baseline) and after the intervention (follow-up). The procedure of this study is shown in Figure 1.

### 2.3. Neuropsychological Assessment

Neuropsychological assessment in this study included general cognitive assessments, the assessment of depression symptoms and detailed neurocognitive function. According to standardized methods, neuropsychologists from a hospital performed the general cognitive assessments on the potential subjects, including the K-MMSE [12,13]. The 15-item Korean version of the Geriatric Depression Scale (SGDS-K) was utilized to measure the subjects’ depression symptoms [19,20]. The Seoul Neuropsychological Screening Battery, Dementia version (SNSB-D) was used to evaluate detailed neurocognitive function [14,15], which includes tests for attention, frontal lobe executive function and verbal functions. The digit span test (forward and backward) was used to assess the function of attention [21]. Moreover, the Korean color-word Stroop test was utilized to assess frontal lobe executive function [14,15]. Finally, word fluency tests (category and letter fluency) were utilized to measure verbal function [22].

### 2.4. Culture-Based VR Training Program

The head-mounted displays consisted of a headset, two controllers and two infrared laser emitter units (HTC Vive, New Taipei City, Taiwan). The HTC Vive used an organic LED display and had an integrated resolution of 2160 × 1200 (1080 × 1200 per eye), with a refresh rate of 90 Hz and a field of view of approximately 110 degrees. The weight of the head-mounted display is approximately 555 g. The head-mounted display had HDMI 1.4, DisplayPort 1.2 and USB 2.0 connections [23].

The VR-based training group underwent training for 30 min/day, two days/week for three months, corresponding to a total 24 times. The VR-based training program was conducted to aid attention, processing speed, executive function and memory conditions, in order to stimulate cognitive function. In each session, 10 min of warm-up and 20 min for two selected games (i.e., memory and/or attention and/or executive function, processing speed) were set for participants according to their level of ability (i.e., high, middle or low), requirements and choice. In VR games, neurologists prepared specific cognitive tasks such as memory function and processing speed while playing games for the elderly. An occupational therapist demonstrated the game controls and gave instructions to ensure that the participants understood how to control the system. The participants were asked to rest and stretch their eyes for a short time when their hypertonicity interfered with the control of the game.

Attention was assessed using the “Crows and Seagulls” game. This game involves shooting crows among seagulls, and helps to improve the level of attention and perceptual space skills. The setting of this game is a traditional bridge in Korea. “Seek a Song of Our Own” is a game that involves participants hitting targets on a screen according to the rhythm and drumming of the “Janggu”, the Korean traditional drum. This game is for elderly participants, and aims to improve the participants’ level of attention. Processing speed was assessed using the “Automated Teller Machine (ATM) machine” game, which involves withdrawing money according to a given amount on the screen. This game aimed to improve numerical ability and perceptivity. To assess frontal lobe executive function, the “Shopping in the Mart” game was used; this game involves shopping in a mart for items needed by the participants’ family. This game aimed to improve the level of perceptivity, numerical ability and logical ability. Both the “ATM machine” and “Shopping in the Mart” games were designed based on activities of daily living. Korean Memory was assessed using the “Fireworks Party” game, which involves decorating a night sky by launching fireworks in a numerical order. This game aimed to improve and train the memory of participants. The “Fruit Cocktail” game was also used to train memory, in which participants made a cocktail by answering questions. Table 1 shows the detailed protocols of the VR-based training program. Figure 2 shows examples of culture-based VR training tasks.

### 2.5. Statistical Analysis

We used IBM SPSS Statistics (version 25.0, SPSS Inc., Chicago, IL, USA) for data analysis in the present study. A Shapiro–Wilk test was used to test the assumptions of distributional normality, and showed that the distribution of all parameters did not differ significantly from normal. The student’s *t*-test was utilized to compare the baseline variables. We also utilized a two-factor repeated-measures analysis of variance (ANOVA) to determine the interaction (group × time) effects for all outcome variables. All results are shown as the mean ± SD. A significant level was set at 0.05.

## 3. Results

The socio-demographic, physical, mental and cognitive baseline characteristics of the participants are shown in Table 2. At baseline, no significant differences were observed in any of the characteristics between the virtual reality (VR) group and the control group. The results also showed that the adherence to the VR-based training program was 91.55% ± 6.41% in the VR group. The only adverse events observed in the treatment groups were dizziness (4.2%) and fatigue (8.3%) during the three-month VR program (data not shown). Table 3 shows the changes in the mental and cognitive function of the VR and control groups at baseline and after 12 weeks. At baseline, no significant differences in the SGDS-K scores between the two groups were found. For the SGDS-K scores, the two-way repeated-measures ANOVA indicated no main effect of time and no interaction between time and group. Within-group analysis indicated that the SGDS-K scores did not change significantly in either group after 12 weeks, but the values tended to be lower than the baseline values in the VR group after 12 weeks (95% CI = −0.87 to 3.07 score, *p* = 0.240). For the K-MMSE scores, the two-way repeated-measures ANOVA indicated no interactions between time and group. Within-group analysis indicated that the K-MMSE scores did not change significantly in either group after 12 weeks. We found no significant changes for both the between-group and within-group analysis of the digit span, Stroop and word fluency test results.

## 4. Discussion

This study fills a research gap, and is the first that uses a clinical setting to examine the effect of culture-based virtual reality (VR) training programs on different domains of cognitive function, and their tolerability in patients with amnestic mild cognitive impairment (aMCI). Our findings showed that the 12-week culture-based VR training program did not improve different domains of cognitive function. Nevertheless, our results further revealed that the culture-based VR training was well-tolerated (negative event of dizziness was 4.2% and fatigue was 8.3%), and with good compliance (compliance rate: 91.55% ± 6.41%) during the three-month VR program in our study sample. Our findings may inform therapists and rehabilitation personnel that culture-based VR training programs can be a promising tool for future training programs for patients with aMCI in the context of culture-based.

Contrary to our hypothesis, our 12-week culture-based VR training program did not improve cognitive function among patients with aMCI compared to the control group. Although existing literature shows that VR is an embodied technology that facilitates cognitive modeling and change by designing virtual environments to simulate both the external and internal world [7], there are several possible explanations for this inconsistency. Firstly, it is inconsistent with a previous study [24] that reported that long-term intervention (six months of VR memory training) in the elderly significantly improved memory function. Moreover, a previous review also reported that short-term VR-based training programs are less effective in improving cognitive function during aMCI or dementia [25]. The short duration of our study might explain this difference in results in patients with aMCI. Another possible explanation could be that our culture-based VR training program was designed to assess attention, executive function and memory conditions in order to stimulate cognitive function, but was not designed for dual tasks (i.e., both cognitive and exercise training). An important review has indicated that exercise plays a critical role in the prevention of dementia [26], and can improve cognitive function in aMCI patients [27]. Previous studies have shown that dual tasks (cognitive and exercise training) could have positive effects on cognitive function among the elderly. For example, Anderson-Hanley et al [28]. found that older adults who participated in stationary cycling with virtual reality tours had improved cognitive function compared to traditional exercisers, which suggested that engaging in both cognitive and physical exercise could be effective for the prevention of cognitive decline [28]. Therefore, future intervention studies may consider using VR-based exercise and cognitive training programs to further confirm the effects of these programs on cognitive function among patients with aMCI.

The present study had some limitations. First, this study is a pilot study, and thus the sample size was small, which limits the statistical power to determine the significance of our findings. Thus, it is still required for further studies using larger sample sizes to identify the effectiveness of regular VR-based training. Second, the time spent in physical activity outside our study was not measured in our participants, and might potentially influence the results of the study. Third, potential confounders such as diet, physical activity, sleep and other cognitive activity were not controlled in this study, and these lifestyle factors, such as physiological and dietary/nutritional profiles [29], as well as physical activity [30] and sleep patterns, might affect our results. Nevertheless, this study using a randomized controlled design could avoid bias, and truly allows for a direct comparison between two groups. Finally, there was only a VR-based intervention group and a control group in this study. Future studies with an ideal active control condition would be needed.

## 5. Conclusions

In conclusion, the present findings demonstrate that the 12-week culture-based virtual reality (VR) training program did not improve different domains of cognitive function during interaction, but the VR-based training was well-tolerated and showed good compliance. Finally, it would be valuable to investigate whether long-term modifications of the culture-based VR training program can help to slow the progression of dementia in a large population.

## Figures and Tables

**Figure 1 ijerph-17-03030-f001:**
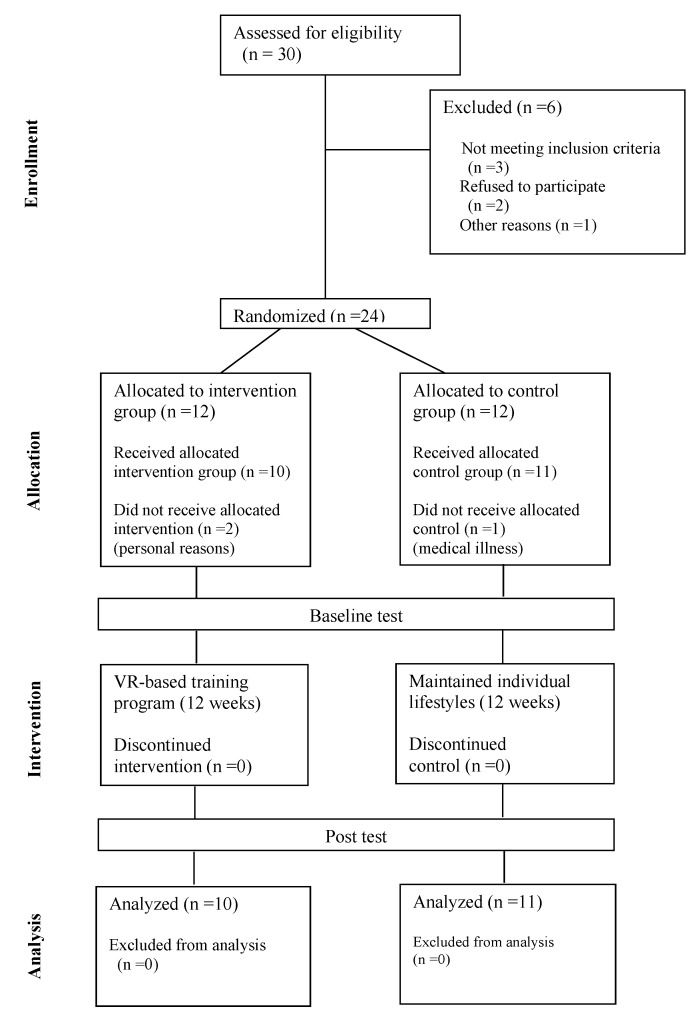
Flowchart from initial contact to study completion. VR: virtual reality.

**Figure 2 ijerph-17-03030-f002:**
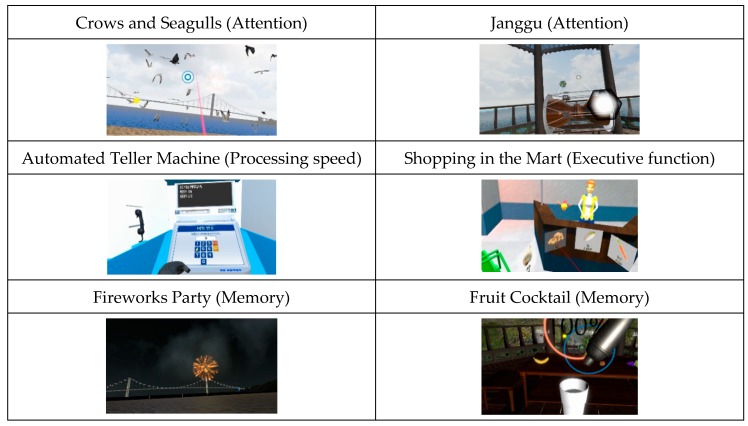
Examples of cultural-based VR training tasks.

**Table 1 ijerph-17-03030-t001:** VR-based training program.

Training Program	Cognitive Domain	Time	Level
Crows and Seagulls	Attention	3–5 min * 2 times	1~4 weeks; Low 5~8 weeks; Middle 9~12 weeks; High
Janggu	Attention	3–5 min * 2 times	
Automated Teller Machine	Processing speed	3–5 min * 2 times	
Shopping in the Mart	Executive function	3–5 min * 2 times	
Fireworks Party	Memory	3–5 min * 2 times	
Fruit Cocktail	Memory	3–5 min * 2 times	

**Table 2 ijerph-17-03030-t002:** Baseline characteristics of socio-demographic, physical, blood pressure, mental and cognitive functions.

Variable	VR Group (*n* = 10)	Control Group (*n* = 11)	*p*-Value
**Socio-Demographic**			
Age (years)	71.80 ± 6.61	69.45 ± 7.45	0.457
Male, *n*(%)	3 (27.3%)	4 (33.3%)	0.772
Educational level (years)	7.20 ± 3.61	8.00 ± 2.90	0.581
Smoking habit (yes), *n*(%)	1 (9.1%)	1 (8.3%)	0.341
Alcohol consumption (yes), *n*(%)	1 (9.1%)	3 (25.0%)	0.331
**Blood Pressure**			
SBP (mmHg)	124.20 ± 23.61	132.72 ± 20.30	0.385
DBP (mmHg)	66.80 ± 11.55	74.73 ± 10.48	0.116
**Mental and Cognitive**			
SGDS-K (score)	6.30 ± 5.03	4.82 ± 4.07	0.465
K-MMSE (score)	25.30 ± 2.41	26.18 ± 1.78	0.348

Values are means ± standard deviations (SD). SBP: systolic blood pressure; DBP: diastolic blood pressure; SGDS-K: Korean version of the Geriatric Depression Scale-Short form; K-MMSE: Korean version of the Mini-Mental State Examination; VR: virtual reality.

**Table 3 ijerph-17-03030-t003:** The changes of mental and cognitive function between the groups at baseline and after 12 weeks.

Variable	Group	Baseline	12 Weeks	Difference	*p*-Value (Interaction)
SGDS-K (score)	VR	6.30 ± 5.03	5.20 ± 4.96	−1.10	0.666
	Control	4.82 ± 4.07	4.27 ± 5.96	−0.55	
K-MMSE (score)	VR	25.30 ± 2.41	25.80 ± 2.74	0.50	0.873
	Control	26.18 ± 1.78	26.82 ± 1.94	0.64	
Digit span forward (score)	VR	6.60 ± 1.26	6.60 ± 1.64	0.00	1.000
	Control	5.91 ± 1.38	5.91 ± 1.30	0.00	
Digit span backward (score)	VR	3.50 ± 1.58	3.60 ± 1.07	1.00	0.637
	Control	3.45 ± 0.69	3.36 ± 1.03	−0.90	
Stroop test: color (score)	VR	60.40 ± 26.81	58.10 ± 35.71	−2.30	0.787
	Control	78.27 ± 30.60	73.91 ± 34.96	−4.36	
Stroop test: word (score)	VR	109.10 ± 5.53	100.60 ± 20.80	−8.50	0.634
	Control	110.91 ± 2.66	105.82 ± 12.34	−5.09	
Word fluency: animal (score)	VR	11.80 ± 2.49	11.60 ± 3.53	−0.20	0.753
	Control	10.64 ± 3.59	11.00 ± 3.79	3.36	
Word fluency: ㄱ (score)	VR	6.70 ± 3.31	8.00 ± 3.27	1.30	0.675
	Control	5.18 ± 2.96	6.00 ± 3.35	0.82	
Word fluency: ㅇ (score)	VR	4.30 ± 2.75	6.20 ± 4.71	1.90	0.231
	Control	4.73 ± 3.32	5.09 ± 2.43	0.36	
Word fluency: ㅅ (score)	VR	6.40 ± 3.47	7.70 ± 4.81	1.30	0.470
	Control	5.18 ± 1.99	5.82 ± 2.48	0.64	

Values are means ± SD. SGDS-K: Korean version of the Geriatric Depression Scale-Short form; K-MMSE: Korean version of Mini-Mental State Examination; and ㄱ, ㅇ and ㅅ: the names of Korean word fluency tests. Significantly different from baseline: *p* < 0.05.

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
