# Peer review of "Feasibility and Tolerability of a Culture-Based Virtual Reality (VR) Training Program in Patients with Mild Cognitive Impairment: A Randomized Controlled Pilot Study"

_ijerph, 2020, doi:10.3390/ijerph17093030_

Round 1
Reviewer 1 Report
The authors conducted a small pilot study of culturally based VR training for adults with aMCI. The VR training targetted a number of cognitive domains over 12 weeks. No differences were found between the active group and controls on neuropsychological assessments at the end of the study period. VR was well tolerated. This is a useful study, and making interventions such as VR culturally bound is an excellent approach. The study has limitations, no least the very small group size and short duration of intervention. These are discussed, but it would be sensible to put more emphasis on these limitations, and also the fact that the control group did not have an active control task. I would not have expected to see significant differences across such small groups, and the main strength of the study is demonstrating feasibility and reporting how they approached the devising challenge of a culturally appropriate VR task.
The tenses used in the sentence structure are not always correct, and the paper requires proof-reading by a person fluent in English grammatical structures.
Author Response
Dear Prof. Dr. Paul B. Tchounwou
Guest Editor,
International Journal of Environmental Research and Public Health
Re: Feasibility and Tolerability of Culture-based Virtual Reality (VR) Training Program in Patients with Mild Cognitive Impairment: A Randomized Controlled Pilot study (Manuscript ID: ijerph-756380)
The authors wish to thank the reviewers for reading our manuscript so thoroughly and providing such constructive feedback. The quality of our manuscript has certainly improved as a result of these comments. Our responses and the necessary changes are included here and within the revised manuscript. We list the comments from each reviewer followed by our responses below. The revised and new sentences are highlighted in yellow background in the revised manuscript.
Responses to the Reviewer 1:
General comment
The authors conducted a small pilot study of culturally based VR training for adults with aMCI. The VR training targetted a number of cognitive domains over 12 weeks. No differences were found between the active group and controls on neuropsychological assessments at the end of the study period. VR was well tolerated. This is a useful study, and making interventions such as VR culturally bound is an excellent approach. The study has limitations, no least the very small group size and short duration of intervention. These are discussed, but it would be sensible to put more emphasis on these limitations, and also the fact that the control group did not have an active control task. I would not have expected to see significant differences across such small groups, and the main strength of the study is demonstrating feasibility and reporting how they approached the devising challenge of a culturally appropriate VR task.
Response: Thank for very much your positive comments.
Query 1: The tenses used in the sentence structure are not always correct, and the paper requires proof-reading by a person fluent in English grammatical structures.
Response1: Thank you very much for your comment. We have sent this manuscript to a English editing company. Please find the edited manuscript in our new submission.
I hope that you find these adjustments satisfactory and that the revised version will be acceptable for publication in the International Journal of Environmental Research and Public Health.
Sincerely yours,
Kyung Won Park

Reviewer 2 Report
Summary
This randomized controlled pilot study examined the feasibility and tolerability of culture-based virtual reality training program in patients with amnestic mild cognitive impairment (aMCI). In conclusion, the authors showed as the culture-based VR training program of 12 weeks did not improve cognitive function during the interaction, the findings revealed that culture-based VR training program was feasible and tolerable for participants with aMCI.
Strengths
The work is generally well written and described, although a further proof-read would be beneficial to catch a few minor grammar errors.
I feel that this would make a valuable contribution to “International Journal of Environmental Research and Public Health (IJERPH)”, and would support its acceptance with “Minor revision”.
Improvements
I think the topic could be interesting and innovative. However, I suggest the following changes or considerations.
Detailed comments
- Line 55:“as shown in Reference[9],” could be changed to “as shown in [9],”.
- In general, authors should decide how to standardize all acronyms in words. Should they be written in lowercase or capital letter? (e.g. line 83-84 - the Seoul Instrumental activities of daily living (IADL).
- I suggest clarifying in the introduction section what the Authors mean “Culture-based”. Please., specify.
- I suggest inserting the number 21 (patients) in the section of the participants. This is to be clear to the reader, as well as excellently shown in figure 1 and the abstract section.
- Line 134: “ 10 min of warm-up and 20 min for two selected games(i.e., memory and/or attention and/or executive function, processing speed)”. How did the authors select the games? it seems as if each patient did specific and individual training ... Please, specify.
- In the discussion section: What is the reason for the dizziness observed in the control group (4.2%)? Please, specify.
Thanks!
Author Response
Dear Prof. Dr. Paul B. Tchounwou
Guest Editor,
International Journal of Environmental Research and Public Health
Re: Feasibility and Tolerability of Culture-based Virtual Reality (VR) Training Program in Patients with Mild Cognitive Impairment: A Randomized Controlled Pilot study (Manuscript ID: ijerph-756380)
The authors wish to thank the reviewers for reading our manuscript so thoroughly and providing such constructive feedback. The quality of our manuscript has certainly improved as a result of these comments. Our responses and the necessary changes are included here and within the revised manuscript. We list the comments from each reviewer followed by our responses below. The revised and new sentences are highlighted in yellow background in the revised manuscript.
Responses to the Reviewer 2:
General Comments
This randomized controlled pilot study examined the feasibility and tolerability of culture-based virtual reality training program in patients with amnestic mild cognitive impairment (aMCI). In conclusion, the authors showed as the culture-based VR training program of 12 weeks did not improve cognitive function during the interaction, the findings revealed that culture-based VR training program was feasible and tolerable for participants with aMCI.
Strengths
The work is generally well written and described, although a further proof-read would be beneficial to catch a few minor grammar errors.
I feel that this would make a valuable contribution to “International Journal of Environmental Research and Public Health (IJERPH)”, and would support its acceptance with “Minor revision”.
Response: Thank you very much for your positive comments.
Improvements
I think the topic could be interesting and innovative. However, I suggest the following changes or considerations.
Query 1: Line 55:“as shown in Reference [9],” could be changed to “as shown in [9],”.
Response 1: Thank you very much for your comments. We have modified this in the revised manuscript. (Page 2, Line 53)
Query 2: In general, authors should decide how to standardize all acronyms in words. Should they be written in lowercase or capital letter? (e.g. line 83-84 - the Seoul Instrumental activities of daily living (IADL).
Response 2: Thank you very much for your comments. We have modified this mistake accordingly. (Page 2, Line 85-86)
Query 3: I suggest clarifying in the introduction section what the Authors mean “Culture-based”. Please., specify.
Response 3: Thank you very much for your comments and suggestions. We have clarified the meaning of “culture-based” in Introduction section. (Page 2, Line 61-63)
For example, a program can be designed based on a Korean traditional game, which may spur older Korean adults’ memory and increase their motivation to attend a VR training program.
Query 4: I suggest inserting the number 21 (patients) in the section of the participants. This is to be clear to the reader, as well as excellently shown in figure 1 and the abstract section.
Response 4: Thank you very much for your suggestions. We have added this important information in the section of Participants (Page 3, Line 98-99).
Finally, subjects were 21 out-patients with aMCI, who were randomized to either the VR-based training group or the control group.
Query 5: Line 134: “10 min of warm-up and 20 min for two selected games (i.e., memory and/or attention and/or executive function, processing speed)”. How did the authors select the games? it seems as if each patient did specific and individual training ... Please, specify.
Response 5: Thank you very much for your comments. Thank you very much for your suggestions. We have added this important information in the section of Culture-based VR Training Program (Page 4, Line 139-141).
In VR games, neurologists prepared specific cognitive tasks, such as memory function and processing speed while playing games for the elderly.
Query 6: In the discussion section: What is the reason for the dizziness observed in the control group (4.2%)? Please, specify.
Response 6: Thank you very much for your comments. We apologize for making this mistake in Abstract section in our original submission. There were no dizziness in control group. The negative events of dizziness (4.2%) were only reported in VR group. Therefore, we have revised this in Abstract section accordingly. (Page 1, Line 28)
The only adverse events observed in the VR group were dizziness (4.2%) and fatigue (8.3%).
I hope that you find these adjustments satisfactory and that the revised version will be acceptable for publication in the International Journal of Environmental Research and Public Health.
Sincerely yours,
Kyung Won Park